# Human Action Recognition: A Paradigm of Best Deep Learning Features Selection and Serial Based Extended Fusion

**DOI:** 10.3390/s21237941

**Published:** 2021-11-28

**Authors:** Seemab Khan, Muhammad Attique Khan, Majed Alhaisoni, Usman Tariq, Hwan-Seung Yong, Ammar Armghan, Fayadh Alenezi

**Affiliations:** 1Department of Computer Science, HITEC University Taxila, Txila 47080, Pakistan; seemab.khan@hitecuni.edu.pk; 2College of Computer Science and Engineering, University of Ha’il, Ha’il 55211, Saudi Arabia; majed.alhaisoni@gmail.com; 3College of Computer Engineering and Science, Prince Sattam Bin Abdulaziz University, Al-Kharaj 11942, Saudi Arabia; u.tariq@psau.edu.sa; 4Department of Computer Science & Engineering, Ewha Womans University, Seoul 120-750, Korea; hsyong@ewha.ac.kr; 5Department of Electrical Engineering, College of Engineering, Jouf University, Sakakah 72311, Saudi Arabia; aarmghan@ju.edu.sa (A.A.); fshenezi@ju.edu.sa (F.A.)

**Keywords:** human action recognition, deep learning, features fusion, features selection, recognition

## Abstract

Human action recognition (HAR) has gained significant attention recently as it can be adopted for a smart surveillance system in Multimedia. However, HAR is a challenging task because of the variety of human actions in daily life. Various solutions based on computer vision (CV) have been proposed in the literature which did not prove to be successful due to large video sequences which need to be processed in surveillance systems. The problem exacerbates in the presence of multi-view cameras. Recently, the development of deep learning (DL)-based systems has shown significant success for HAR even for multi-view camera systems. In this research work, a DL-based design is proposed for HAR. The proposed design consists of multiple steps including feature mapping, feature fusion and feature selection. For the initial feature mapping step, two pre-trained models are considered, such as DenseNet201 and InceptionV3. Later, the extracted deep features are fused using the Serial based Extended (SbE) approach. Later on, the best features are selected using Kurtosis-controlled Weighted KNN. The selected features are classified using several supervised learning algorithms. To show the efficacy of the proposed design, we used several datasets, such as KTH, IXMAS, WVU, and Hollywood. Experimental results showed that the proposed design achieved accuracies of 99.3%, 97.4%, 99.8%, and 99.9%, respectively, on these datasets. Furthermore, the feature selection step performed better in terms of computational time compared with the state-of-the-art.

## 1. Introduction

Human action recognition (HAR) emerged as an active research area in the field of computer vision (CV) in the last decade [1]. HAR has applications in various domains including; surveillance [2], human-computer interaction (HCI) [3], video reclamation, and understanding of visual information [4], etc. The most important application of action recognition is video surveillance [5]. Governments use this application for intelligence gathering, reducing crime rate, for security purposes [6], or even crime investigation [7]. The main motivation of growing research in HAR is due to its use in video surveillance applications [8]. In visual surveillance, HAR plays a key role in recognizing the activities of subjects in public places. Furthermore, these types of systems are also useful in smart cities surveillance [9].

Human actions are of various types. These actions can be categorized into two broad classes, namely voluntary actions and involuntary actions [10]. Manual recognition of these actions in real-time is a tedious and error-prone task; therefore, many CV techniques are introduced in the literature [11,12] to serve this task. Most of the proposed solutions are based on classical techniques such as shape features, texture features, point features, and geometric features [13]. A few techniques are based on the temporal information of the human [14], and a few of them extract human silhouettes before feature extraction [15].

Recently, deep learning has shown promising results in the field of computer vision (CV) [16]. Deep learning makes learning and data representation at multiple levels by mimicking the human brain processing [17] to create models. These models consist of multiple processing layers such as convolutional, ReLu, pooling, fully connected, and Softmax [18]. The functionality of a CNN model is to replicate the working of the human brain as it preserves and makes sense of multidimensional information. There exist multiple methods in deep learning, which include encompassing neural networks, hierarchical probabilistic models, supervised learning, and unsupervised learning models [19].

The HAR process is a challenging task as there are a variety of human actions in daily life. In order to tackle this challenge, deep learning models are utilized. The performance of a deep learning model is always based on the number of training samples [20]. In the action recognition tasks, several datasets are publicly available. These datasets include several actions such as walking, running, leaving a car, waving, kicking, boxing, throwing, falling, bending down, and many more.

Recently proposed systems mainly focus on the hybrid techniques; however, they do not focus on minimizing the computational time [21]. This is an important factor as most time surveillance is performed in real-time. Some of the other key challenges of HAR are as follows: (i) Query video sequences resolution is imperative for the recognition of the focal point in the most recent frame. The background complexity, shadows, lighting conditions, and outfit conditions extract irrelevant information using classical techniques of human action, which later results in inefficient action classification; (ii) with automatic activities recognition under multi-view cameras it is difficult to classify the correct human activities. Change in the motion variation captures the wrong activities under the multi-view cameras; (iii) imbalanced datasets impact the learning of a CNN. A CNN model always needs a massive number of training images for learning; and (iv) features extraction from the entire video sequences includes several irrelevant features, affecting the classification accuracy.

These challenges are considered in this work to propose a fully automated design using deep learning features fusion and best feature selection for HAR under the complex video sequences. The major contributions of this work are summarized as follows:Selected two pre-trained deep learning models and removed the last three layers. The new layers are added and trained on the target datasets (action recognition dataset). In the training process, the first 80% of the layers are frozen instead of using all the layers, whereas the training process was conducted using transfer learning.Proposed a Serial based Extended (SbE) approach for multiple deep learning features fusion. This approach fused features in two phases for better performance and to reduce redundancy.Proposed a feature selection technique named Kurtosis-controlled Weighted KNN (KcWKNN). A threshold function is defined which is further analyzed using a fitness function.Performed an ablation study to investigate the performance of each step in terms of advantages and disadvantages.

The rest of the manuscript is organized as follows: Related work is presenting in Section 2. The proposed design for HAR is presented in Section 3, which includes deep learning models, transfer learning, the selection of best features and fusion. Results of the proposed method are presented in Section 4 in terms of tables and confusion matrixes. Finally, Section 5 concludes this work.

## 2. Related Work

HAR has emerged as an impactful research area in CV from the last decade [22]. It is based on important applications such as visual surveillance [23], robotics, biometrics [24,25], and smart healthcare centers to name a few [26,27]. Several researchers of computer vision developed techniques using machine learning [28] for HAR. Most of these researches focused on deep learning due to its better performance and few of them used barometric sensors for activity recognition [29]. Rasel et al. [30] extracted the spatial features using acidometer sensors and classified using multiclass SVM for final activity recognition. Zhao et al. [31] introduced a combined framework for activity recognition. They combined short-term and long-term features for the final results. Khan et al. [32] combined the attention-based LSTM network with dilated CNN model features for the action recognition. Similarly, a skeleton based attention framework is presented by [33] for action recognition. Maheshkumar et al. [13] presented an HAR framework using both the shape and the OFF features [34]. The presented framework is the combination of Hidden Markov Model (HMM) and SVM. The shape and OFF features are extracted and used for HAR through the HMM classifier. The multi-frame averaging method was adopted for background extraction of the image. A discrete Fourier transform (DFT) was performed to reduce the magnitude on the length feature set from the middle to the body contour. In order to select features, the principal component analysis was implied. The presented framework was tested on videos recorded in real-time settings and achieved maximum accuracy. Weifeng et al. [35] presented a generalized Laplacian Regularized Sparse Coding (LRSC) framework for HAR. It was a nonlinear generalized version of graph Laplacian with a tighter isoperimetric inequality. A fast-iterative shrinkage thresholding algorithm for the optimization of ρ-LRSC was also presented in this work. The input of the sparse codes learned by the ρ-LRSC algorithm were placed into the support vector machine (SVM) for final categorization. The datasets used for the experimental process were unstructured social activity attribute (USAA) and HMDB51. The experimental results demonstrated the competence of the presented ρ-LRSC algorithm. Ahmed et al. [36] presented an HAR model using a depth video analysis. HMM was employed to recognize regular activities of aged people living without any attendant. The first step was to analyze the depth maps through the temporal motion identification method using the segments of human silhouettes in a given scenario. Robust features were selected and fused together to find the gradient orientation change, intensity difference temporal and local movement of the body organs [37]. These fused features were processed via embedded HMM. The experimental process was conducted on three different datasets such as Online Self-Annotated [38], Smart Home, and Three Healthcare, and achieved the accuracies 84.4, 87.3, and 95.97%, respectively. Muhammed et al. [39] presented a smartphone inertial sensors-based framework for human activity recognition. The presented framework was divided into three steps: (i) extract the efficient features; (ii) the features were reduced using the kernel principal component analysis (KPCA) and linear discriminant analysis (LDA) to make them resilient; (iii) resultant features were trained via deep belief neural networks (DBN) to attain improved accuracy. The presented approach was compared with traditional expression recognition approaches such as typical multiclass SVM [40,41] and artificial neural network (ANN) and showed an improved accuracy.

Lei et al. [42] presented a light weight action recognition framework based on DNN using RGB video sequences. The presented framework was constructed using CNNs and LTSM units that was a temporal attention model. The purpose of using CNNs was to segment out the objects from the complex background. LTSM networks were used on spatial feature maps of multiple CNN layers. Three datasets, such as UCF-11, UCF Sports, and UCF-101, were used for experimental processes and achieved 98.45%, 91.89%, and 84.10%, respectively. Abdu et al. [43] presented an HAR framework based on deep learning. They considered the problem of traditional techniques which are not useful for the better accuracy of complex activities. The presented framework used a cross DBNN model that unites the SRUs with GRUs of the neural network. The SRUs were used to execute the sequence multi-modal data input. Then GRUs were used to store and learn the amount of information that can be transferred from past state to future state. Zan et al. [44] presented an action recognition model that served the problem of multi-view HAR. The presented algorithm was based on adaptive fusion and category-level dictionary learning (AFCDL). In order to integrate dictionary learning, query sets were designed, and the regularization scheme was constructed for the adaptive weights assignment. Muhammad et al. [45] presented a new framework of 26-layered CNN for composite action classification. Two layers, the global average pooling layer and fully connected layer (FC) were used for feature extraction. The extracted features are classified using the extreme learning machine (ELM) and Softmax for final action classification. Four datasets named HMDB51, UCF Sports, KTH, and Weizmann were used for the experimentation process and showed better performance. Muhammad et al. [4] presented a new fully automated structure for HAR by fusing DNN and multi-view features. Initially, a pre-trained CNN named VGG19 was implied to take out DNN features. Horizontal and vertical gradients were used to compute multi-view features and vertical directional attributes. Final recognition was performed on the selected features via the Naive Bayes Classifier (NBC). Kun et al. [46] introduced an HAR model based on DNN that combines the convolutional layer with LSTM. The presented model was able to automatically extract the features and perform their classification with the standard parameters.

Recently, the development of deep learning models for HAR using high dimensional datasets has shown immense progress. Classical methods for HAR did not show satisfactory performance, especially for large datasets. In contrast, the modern techniques such as Long Short-Term Memory (LSTM), SV-GCN, and Convolution Neural Networks (CNNs) are showing improved performance and can be considered for further research to obtain an improvement in the accuracy.

## 3. Proposed Methodology

This section presents the proposed methodology for human action recognition in complex video sequences. The proposed design consists of multiple steps, including feature mapping, feature fusion, and feature selection. Figure 1 represents the proposed design of HAR. In this design, features are extracted from the two pre-trained models such as DenseNet201 and InceptionV3. The extracted deep features are fused using the Serial based Extended (SbE) approach. In the later step, the best features are selected using Kurtosis-controlled Weighted KNN. The selected features are classified using several supervised learning algorithms. Detail of each step is provided below.

### 3.1. Convolutional Neural Network (CNN)

CNN is an innovative technique in deep learning that makes the classification process fast and precise. CNN requires lesser parameters to train compared with the traditional neural networks [47]. A CNN model contains multiple layers where the convolution layer is an integral part. Few other layers contained in the CNN model are pooling layers (min, max, average), the ReLU layer, and some fully connected (FC) layers. The internal structure of a CNN has multiple layers as presented in Figure 2. This figure shows that video sequences are provided as input to this network. In the network, the initially convolutional layer is added to convolve input image features, which are later normalized in pooling and hidden layers. After that, FC layers are added to convert image features into 1D feature vector. The final 1D extracted features are classified in the last layer, which is known as the output layer.

### 3.2. Densenet201 Pre-Trained Deep Model

DenseNet is an advanced CNN model where every layer is directly connected with all the layers in subsequent order. These connections help to improve the flow of information in the network, as illustrated in Figure 3. This dense connectivity makes it a dense convolutional network commonly known as DenseNet [48]. Other than the improvement in the information flow, it caters to the vanishing gradient problems as well as it strengthens the feature prorogation process. DenseNet also allows for reusing the features and it reduces required parameters, which eventually reduces the computational complexity of the algorithm. Consider a CNN with ϕ number of layers and $\phi_{l}$ layer index has an input stream that starts with $x_{0}$. A nonlinear transformation function $F_{\phi }\left(.\right)$ is applied on each layer and it can be a combination of multiple functions such as BN, pooling convolution or ReLU. In a densely connected network, each layer is connected to its subsequent layers. Output of the $\phi^{th}$ layer is represented by $x_{\phi }$.
(1)$$x_{\phi }=F_{\phi }\left(x_{0},\ldots \ldots ,x_{\phi -1}\right)$$
where $\left(x_{0},\ldots \ldots ,x_{\phi -1}\right)$ states the concatenation of the feature maps generated in layers 0,……..,ϕ−1.

### 3.3. Inception V3 Pre-Trained Deep Model

InceptionV3 [49] is an already trained CNN model on the ImageNet dataset. It consists of 316 layers which include convolution layers, pooling layers, fully connected layers, dropout, and Softmax layers. The total number of connections in this model is 350. Unlike a traditional CNN that allows a fixed filter size in a single layer, InceptionV3 has the flexibility to use variable filter sizes and a number of parameters in a single layer which results in better performance. An architecture of InceptionV3 is shown in Figure 4.

### 3.4. Transfer Learning Based Learning

Transfer learning is a well-known technique in the field of deep learning that allows the reusability of a pre-trained model on an advanced research problem [50]. A major advantage of using TL is that it requires less data as input and provides remarkable results. It aims to transfer knowledge from a source domain to a targeted domain, here the source domain refers to a pre-trained model with a very large dataset and the targeted domain is the proposed problem with limited labels [51]. In the source domain, usually a large high-resolution image dataset known as ImageNet is used [52,53]. It contains more than 15 billion labels and 1000 image categories. Image labels in ImageNet are saved according to the wordNet hierarchy, where each node leads to thousands of images belonging to that category. Mathematically, TL is defined as follows:

Given a source domain $s_{d}$, defined as:$$s_{d}=\left\{\left(x_{1}^{d},y_{1}^{d}\right),\;\ldots ,\left(x_{i}^{d},y_{i}^{d}\right),\ldots ,\;\left(x_{n}^{d},y_{n}^{d}\right)\right\}$$

The learning task is $L_{d}$,$L_{s}$,$\left(x_{m}^{d},y_{m}^{d}\right)$ ∈ φ. The target domain is defined as:$$s_{t}=\left\{\left(x_{1}^{t},y_{1}^{t}\right),\;\ldots ,\left(x_{i}^{t},y_{i}^{t}\right),\ldots ,\;\left(x_{n}^{t},y_{n}^{t}\right)\right\}$$

The learning task $L_{t}$, $\left(x_{n}^{t},y_{n}^{t}\right)$ ∈φ, (m,n) will be the size of training data, where n≪m and $y_{i}^{d}\;\mathrm{and}\;y_{i}^{t}$ are the training data labels. Using this definition, both pre-trained models are trained on action datasets. During the training process, the learning rate was 0.01, the mini batch size is 64, the maximum epochs is 100 and the learning method is the stochastic gradient descent. After the fine-tuning process, the output of both models is the number of action classes.

### 3.5. Features Extraction

Features are extracted from the newly learned models called target models as shown in Figure 5 and Figure 6. Figure 5 represents a DenseNet201 modified model. Using this model, features are extracted using the avg-pool layer. In the output, an N×1920 dimensional feature vector was obtained, denoted by $\vec{C}$, where N represents number of images in the target dataset.

Using the Inception V3 modified model (depicted in Figure 6), features are extracted from the average pool layer. On this layer, the dimension of the extracted deep feature vector is N×2048 and it is represented by $\vec{D}$, where N is the number of images in the target dataset.

### 3.6. Serial Based Extended Fusion

The fusion of features is becoming a popular technique for improved classification results. The main advantage of this step is to improve the image information in terms of features. The improved feature space increases the classification performance. In the proposed work, a Serial based Extended (SbE) approach is implemented. In this approach, initially features are fused using a serial-based approach. The fused vectors are combined in a single feature vector and to obtain a feature vector of dimension N×3968 and denoted by ƥ, considering two feature vectors C→ and D→ defined on the outline of sample space Z→. For an arbitrary sample ᵹ∈Z→, the equivalent two feature vectors are j∈ C→ and k∈ D→. The serial combined feature of ᵹ can be defined as ƥ=(jk). If feature vector C→ has n dimensions and feature vector D→ has m dimensions, then serial fused feature ƥ will have (n+m) dimensions. After obtaining a ƥ feature vector, the features are sorted into descending order and the mean value is computed. Based on the mean value, the feature vector is extended in terms of the final fusion.
(2)μ(ƥ)=1ℕ∑i=1ℕ(ƥi)
(3)Fsn={Fusion(i)  for ƥi≥μDiscard,   ElseWhere

Here, Fusion(i) is a final fused feature vector of dimension N×*K*, where the value of K is always transformed according to the variation in the dataset. Later on, this fusion vector is analyzed using the experimental process and further refined using a feature selection approach.

### 3.7. Serial Based Extended Fusion

Feature selection is the process of the selection of subset features from the input feature vector [54]. It helps to improve the performance of the algorithm and also reduces the training time. In the proposed design, a new feature selection algorithm is proposed, Kurtosis-controlled Weighted KNN (KcWKNN). The proposed selection method works in the following steps: (i) input fused feature vector; (ii) compute Kurtosis value; (iii) define a threshold function; (iv) calculate fitness, and (v) select the feature vector.

The Kurtosis value is computed as follows:(4)$$Kr=\frac{\mu_{4}}{\delta^{4}}$$
(5)$$\mu_{4}=E\left[{\left({\overbrace{F}}_{i}-E\left[\overbrace{F}\right]\right)}^{n}\right],\;{\overbrace{F}}_{i}\in Fusion\left(i\right)\;\mathrm{and}\;n=4$$
(6)$$\delta^{4}=\sqrt{E\left[{\left({\overbrace{F}}_{i}-\mu \right)}^{2}\right]}$$
where K is the Kurtosis function, $\mu_{4}$ is the fourth central moment, and δ is the standard deviation. Kurtosis is a statistical measure that we investigate to find how much the tails of the distribution deviate from the normal. Distributions with higher values are identified in this process. In this work, the main purpose of using Kurtosis is to obtain the higher tail values (outlier features) through the fourth moment that was later employed in the threshold function for the initial feature selection. By using the Kurtosis value, a threshold function is defined as follows:(7)$$Ts=\{\begin{array}{c} FS\left(i\right)\;\;\;\;for\;\;\;Fusion\left(i\right)\geq Kr\\ Ignore,\;\;\;\;\;\;\;\;\;\;\;\;\;\;\;Elsewhere\; \end{array}$$

The selected feature vector FS(i) is passed into the fitness function WKNN for validation. Mathematically, WKNN is defined as follows:

Consider ${\left\{\left(x_{i},y_{i}\right)\right\}}_{i=1\;}^{N}\in \;Ρ$ as the training set where $x_{i}$ is the *p*-dimensional training vector and $y_{i}$ is its equivalent class labels set. To determine the label $\bar{y}$ of any $\bar{x}$ from the test set $\left(\bar{x},\;\bar{y}\right)$, the following mathematics takes place.

(a)Compute the Euclidian distance e between $\bar{x}$ and each $\left(\bar{x},\;\bar{y}\right)$, formal given in Equation (8).
(8)$$e\left(\bar{x},\;\;x_{i}\right)=\bar{x}-x_{i}{}_{i_{o}}$$(b)Arrange all values in ascending order(c)Assign a weight ${ώ}_{i}$ to the ith nearest neighbor using Equation (9).
(9)$${ώ}_{i}=\frac{1}{{\left(e\left(\bar{x},\;\;x_{i}\right)\right)}^{2}}$$(d)Assign ${ώ}_{i}=1$ for the equally weighted KNN rule,(e)The class label of $\bar{x}\;$
is assigned on the basis of majority votes from the neighbors by Equation (10).
(10)$$\bar{y}=\arg\max\sum_{\left(x,y\right)\in Ρ}{ώ}_{i},\;ᶯ\left(x={\bar{y}}_{i}\right)$$
where x is the class label, ${\bar{y}}_{i}$ is the class label for ith nearest neighbor and ᶯ(.) is the Dirac-Delta function that takes value = 1 if its argument is true and 0 otherwise.(f)Compute error.

The error is used as a performance measure, where the number of iterations is initialized as 50. This process is carried out until the error is minimized. Visually, the flow is shown in Figure 7, where it can be seen that the best selected features are finally classified using supervised learning algorithms. Moreover, the complete work of the proposed design is listed in Algorithm 1.
**Algorithm 1.** The complete work of the proposed design. **Input:** Action Recognition Datasets**Output:** Predicted Action ClassStep 1: Input action datasetsStep 2: Load Pre-trained Deep Models;-Densenet201-Inception V3Step 3: Fine Deep ModelsStep 4: Trained Deep Models using TLStep 5: Feature Extraction from Avg Pooling LayersStep 6: SbE approach for Features FusionStep 7: Best Features Selection using Proposed KcWKNNStep 8: Predict Action Label

## 4. Results and Analysis

The experimental process of the proposed method is presented in this section. Four publically available datasets such as KTH [3], Hollywood [38], WVU [39], and IXMAS [40] were used in this work for the experimental process. Each class of these datasets contains 10,000 video frames that are utilized for the experimental process. In the experimental process, 50% of video sequences are used for the training purpose, while the remaining 50% is utilized for the testing purpose. The K-Fold cross validation is adopted, where the value of *K* = 10. Results are computed on several supervised learning algorithms and select the best one is selected based on the accuracy value. All simulations are conducted on MATLAB2020a using a Personal Computer Corei7 with 16 GB of RAM and 8 GB Graphics card.

### 4.1. Results

A total of four experiments were performed on each dataset to analyze the performance of the middle step. These steps are: (i) performed classification using DenseNet201 deep features; (ii) performed classification using InceptionV3 deep model; (iii) performed classification using the SbE deep features fusion, and (iv) performed classification using KcWKNN-based feature selection.

**Experiment 1:** Table 1 presents the results of the specific DenseNet201 deep features on selected datasets. In this table, it is noted that the Cubic SVM achieved a better accuracy of 99.3% on the KTH dataset. Other classifiers also achieved a better accuracy of above 94%. For the Hollywood action dataset, the best achieved accuracy is 99.9% for Fine KNN. Similar to the KTH dataset, the rest of the classifiers also performed better on this dataset. The best obtained accuracy for the WVU dataset is 99.8% for Cubic SVM. The rest of the classifiers also performed better and achieved an average accuracy of 97%. The best obtained accuracy of the IXAMAS dataset is 97.3% for Fine KNN.

**Experiment 2:** The results of InceptionV3 deep features are provided in Table 2. In this table, it is noted that the best achieved accuracy on the KTH dataset is 98.1%, for the Hollywood dataset it is 99.8%, for the WVU dataset it is 99.1%, and for the IXAMAS dataset it is 96%. From this table, it is observed that the performance of specific DenseNet201 features are better. However, during the computation of results, time significantly increases. Therefore, it is essential to handle this issue with consistent accuracy.

**Experiment 3:** After the experiments on specific feature sets, the SbE approach is applied for deep features fusion. The KTH dataset results are provided in Table 3. In this table, The highest performance is recorded for Cubic SVM with an accuracy of 99.3%. Recall and precision are 99.3% and 99.43% respectively. Moreover, the noted time during the training process is 893.23 s. The second highest accuracy is achieved by a linear discriminant classifier of 99.2%. The rest of the classifiers also performed better. Compared with specific feature vectors, the fusion process results are more consistent. Figure 8 illustrates the true positive rates (TPRs)-based confusion matrix of Cubic SVM that confirms the value of the recall rate. In this figure, the highlighted diagonal values represent the true positive predictions, whereas the values other than the diagonal represent false negative predictions.

Table 4 represents the results of the Hollywood action dataset using the SbE approach. In this table, it is noted that the best accuracy is 99.9%, obtained by Fine KNN. Other performance measures such as recall rate, precision rate and F1 score values are 99.1825%, 99.8375%, and 99.5089%, respectively. The rest of the classifiers mentioned in this table performed better and achieved an average accuracy above 98%. Figure 9 illustrates the TPR-based confusion matrix of Fine KNN, where it is clear that each class prediction rate is above 99%. Moreover, compared with the specific deep features experiment on the Hollywood dataset, the fusion process shows more consistent results.

Table 5 presents the results of the WVU dataset using the SbE fusion approach. The highest accuracy is achieved through Linear Discriminant which is 99.8%, where the recall rate, precision rate, and F1 score are 99.79%, 99.78%, and 99.78%, respectively. Quadratic SVM and Cubic SVM performed second best and achieved an accuracy of 99.7% for each. The rest of the classifiers also performed better and gained the average accuracy of above 99%. Figure 10 illustrated the TPR based confusion matrix of the WVU dataset for the Linear Discriminant classifier. This figure showed that the correct prediction rate of each classifier is more than 99%. Compared with this accuracy of WVU on specific features, it is noticed that the fusion process provides consistent accuracy.

Table 6 presents the results of the IXMAS dataset after SbE features fusion. In this table, it can be seen that the highest accuracy is achieved through Fine KNN of 97.4%, where the recall rate, precision rate, and F1 score are 97.18%, 97.25%, and 97.21%, respectively. Cubic SVM performed second best and achieved an accuracy of 97.3%. The rest of the classifiers also performed better and attained an average accuracy above 93%. Figure 11 illustrates the TPR-based confusion matrix of the Fine KNN for the IXMAS dataset using the SbE approach.

Overall, the results of the SbE approach are improved and are consistent compared with the specific deep features (see results in Table 1 and Table 2). However, it is observed that the computational time increases during the fusion process. For a real-time system, this time needs to be minimized. Therefore, a feature selection approach is proposed.

**Experiment 4:** In this experiment, the best features are selected using Kurtosis-controlled WKNN and provided to the classifiers. Results are provided in Table 7, Table 8, Table 9 and Table 10. Table 7 presents the results of the proposed feature selection algorithm on the KTH dataset. In this table, the highest obtained accuracy is 99%, achieved by Cubic SVM. Other performance measures such as recall, precision and F1 score are 98.1666%, 99.1166% and 99.016%, respectively. Figure 12 illustrates the TPR-based confusion matrix of the Cubic SVM for the best feature selection process. In comparison with Table 3 results, it is noted that the accuracy of Cubic SVM decreases (0.3%), while the computational time expressively declines. The computational time of the Cubic SVM in the fusion process was 893.23 s, which is reduced after the feature selection process to 451.40 s. This shows that the feature selection process not only maintains the recognition accuracy but also minimizes the computational time.

Table 8 presents the best feature selection results on the Hollywood Action dataset and achieved best accuracy by Fine KNN of 99.8%. The other calculated measures such as recall rate, precision rate, and F1 Score are 99.812%, 99.837%, and 99.82%, respectively. For the rest of the classifiers, the average accuracy is above 98% (can be seen in this table). Figure 13 illustrates the TPR-based confusion matrix of Fine KNN for this experiment. The diagonal values in this experiment show the correct predicted values. Comparison with Table 4 shows that the classification accuracy is still consistent, whereas the computational time is significantly reduced. The computational time at the fusion process was 447.76 s, whereas after the selection process, it is reduced to 213.33 s. This shows that the selection of best features using KcWKNN performed significantly better.

Table 9 presents the results of the WVU dataset after the best feature selection using KcWKK. In this table, Quadratic SVM and Cubic SVM performed best with the accuracy of 99.4%, where the recall rate is 99.37% and 99.43%, respectively, and the precision rate is 99.38% and 99.44%, respectively and the F1 score is 99.375%, and 99.43%, respectively. Figure 14 shows the TPR-based confusion matrix of the Cubic SVM for this experiment. This figure shows that the prediction rate of each class is above 99%. Moreover, in comparison with Table 5 (fusion results), the computational time of this experiment on the WVU dataset is almost half and accuracy is still consistent. This shows that the KcWKNN selection approach performed significantly well.

The results of the KcWKNN-based best features selection on the IXMAS dataset are provided in Table 10. In this table, it is noted that the Fine KNN attained best accuracy of 97.1%, whereas the recall rate, precision rate, and F1 score are 97.075%, 96.9916%, and 97.033%, respectively. Figure 15 illustrates the TPR-based confusion matrix of the Fine KNN for this experiment. The correct prediction value of each class is provided in the diagonal of this figure. Compared with Table 6, this experiment reduces the computational time while maintaining the recognition accuracy.

Finally, a detailed analysis was conducted among all experiments in terms of accuracy and time. From Table 1, Table 2, Table 3, Table 4, Table 5, Table 6, Table 7, Table 8, Table 9 and Table 10, it is observed that the accuracy value is improved after the proposed fusion process and the time is reduced. However, the noted time was still high and must be reduced further; therefore, a feature selection technique is proposed and time is significantly reduced compared with the original extracted deep features and fusion step (plotted in Figure 16, Figure 17, Figure 18 and Figure 19). In the selection process, a little change occurred in the accuracy value, but on the other side, a high fall is noted in the computational time.

### 4.2. Comparison with SOTA

Overall, the feature selection process maintains the classification accuracy while significantly reducing the computational time. A comparison with some recent techniques was also conducted as provided in Table 11. This table shows that the proposed design results are significantly improved. The main strength of the proposed design is the fusion of deep features using the SbE approach and best feature selection using KcWKNN.

## 5. Conclusions

HAR has gained a lot of popularity in recent years. Multiple techniques have been used for the accurate recognition of human actions. The problem is to correctly identify the action in real-time and from multiple perspectives. In this work, a design is proposed where the key aim is to improve the accuracy of the HAR process in the complex video sequences using advanced deep learning techniques. The proposed design consists of four steps, namely feature mapping, feature fusion, feature selection, and classification. Two modified deep learning models, DenseNet201 and InceptionV3 were used for feature mapping. Fusion and selection were performed using the serial-based extended approach and Kurtosis-controlled Weighted KNN approach, respectively. The results were obtained after extensive experimentation on state-of-the-art action datasets. Based on the results, it is concluded that the proposed design performed better than the existing techniques in terms of accuracy as well as computational time. Cubic SVM and Fine KNN classifiers were top performers on the proposed HAR method. The key limitation of this work is the computational time that was noted during the original deep extracted features. This step increases the computational time that is not suitable for the real-time applications. As a future study, we intend to test the proposed design on relatively complex action datasets such as HMDB51 and UCF101. Moreover, the recent deep learning models can also be considered for feature extraction and will study the less complexity feature fusion and selection algorithms.

## Figures and Tables

**Figure 1 sensors-21-07941-f001:**
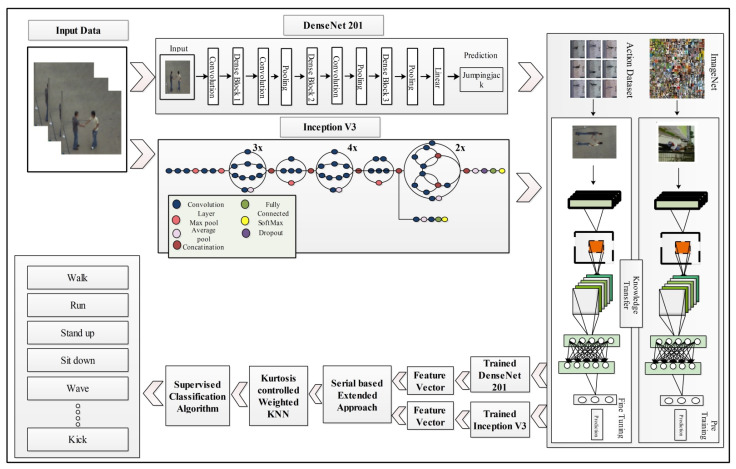
Illustration of a proposed design for HAR using deep learning.

**Figure 2 sensors-21-07941-f002:**
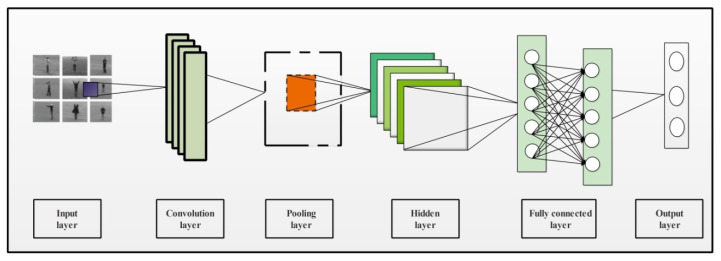
A simple architecture of CNN containing multiple layers for image classification.

**Figure 3 sensors-21-07941-f003:**
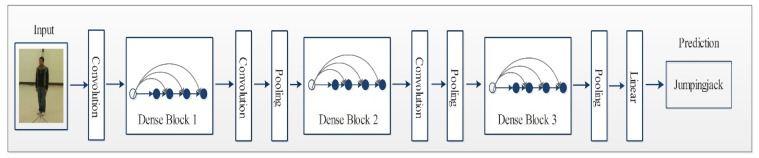
Network architecture of DenseNet201 for action recognition.

**Figure 4 sensors-21-07941-f004:**
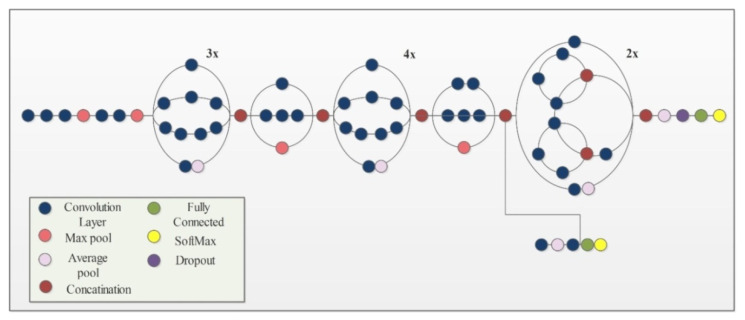
Network architecture of Inceptionv3 model.

**Figure 5 sensors-21-07941-f005:**
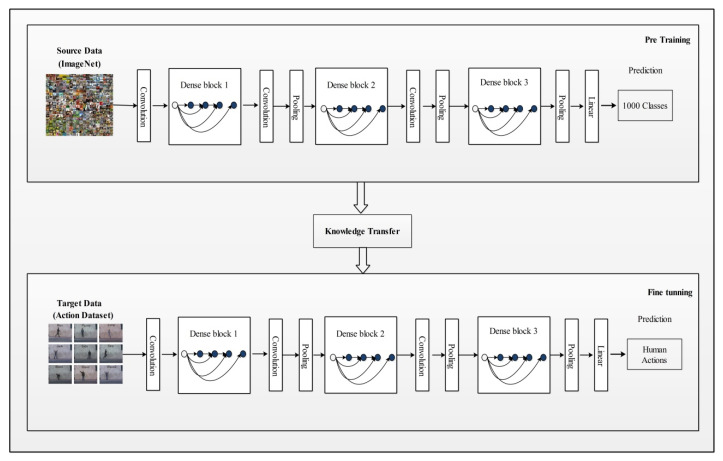
Target model (modified DenseNet201) for feature extraction.

**Figure 6 sensors-21-07941-f006:**
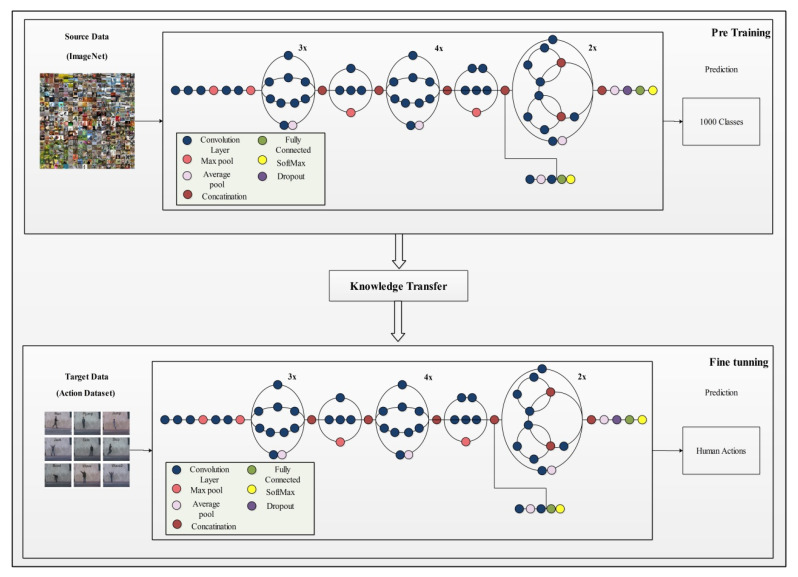
Target model (modified Inception V3) for feature extraction.

**Figure 7 sensors-21-07941-f007:**
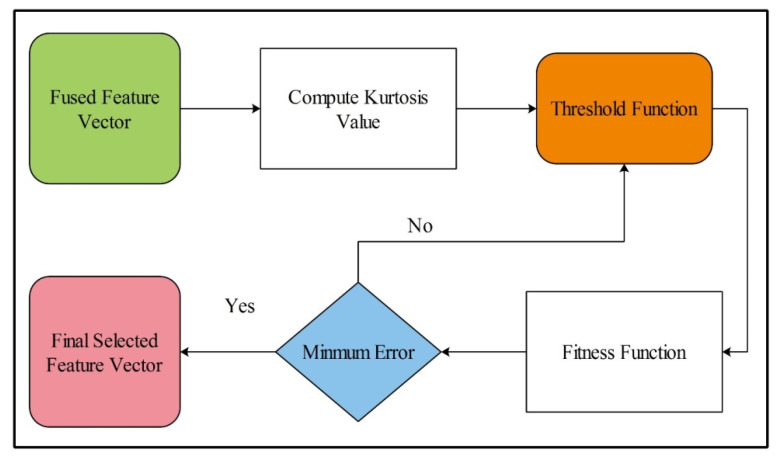
Proposed flow diagram of best feature selection.

**Figure 8 sensors-21-07941-f008:**
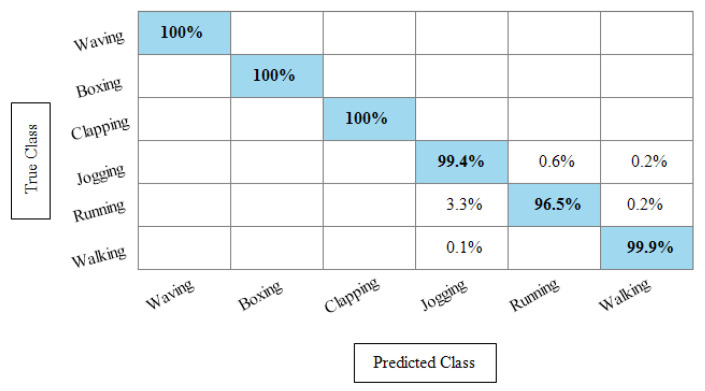
TPR-based confusion matrix of KTH dataset after fusion of deep features using SbE approach.

**Figure 9 sensors-21-07941-f009:**
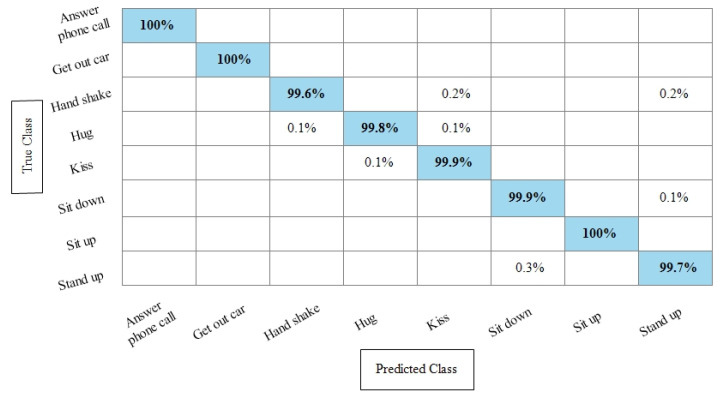
TPR based confusion matrix of Fine KNN using Hollywood dataset after fusion of deep features through SbE approach.

**Figure 10 sensors-21-07941-f010:**
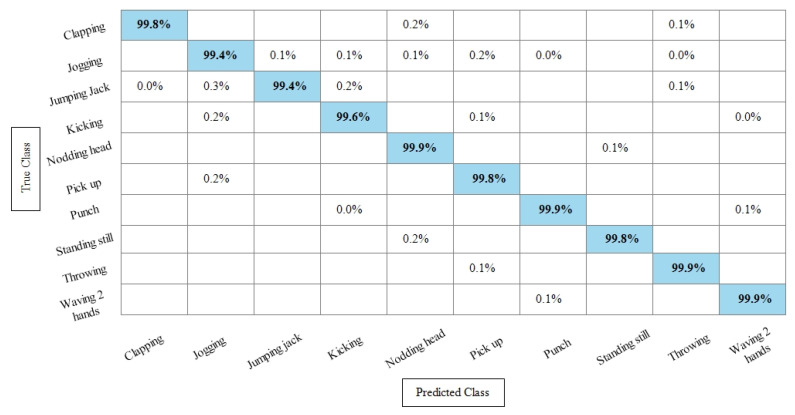
TPR-based confusion matrix of Linear Discriminant classifier after fusion of deep features using SbE approach.

**Figure 11 sensors-21-07941-f011:**
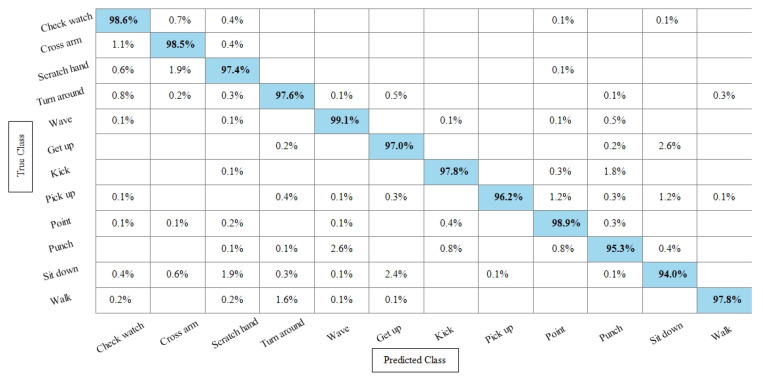
TPR-based confusion matrix of Fine KNN after fusion of deep features using SbE approach.

**Figure 12 sensors-21-07941-f012:**
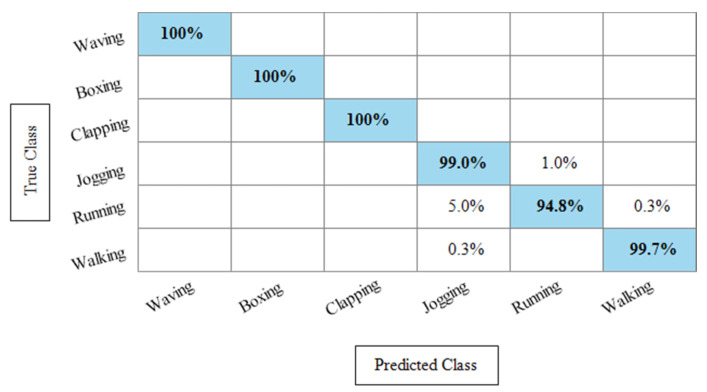
TPR based confusion matrix of Cubic SVM after best feature selection using KcWKNN.

**Figure 13 sensors-21-07941-f013:**
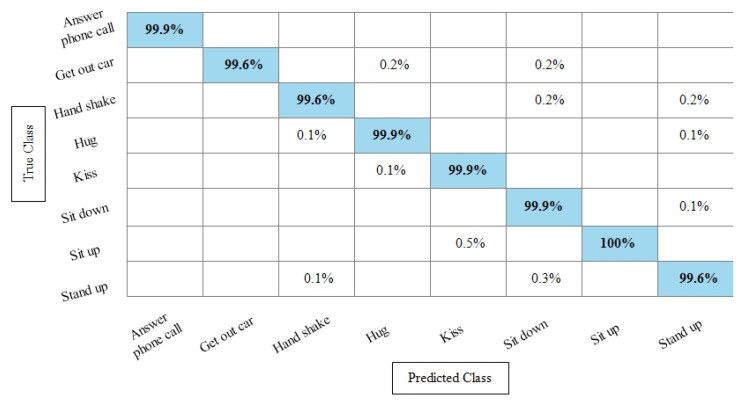
TPR based confusion matrix of Fine KNN after best feature selection using KcWKNN.

**Figure 14 sensors-21-07941-f014:**
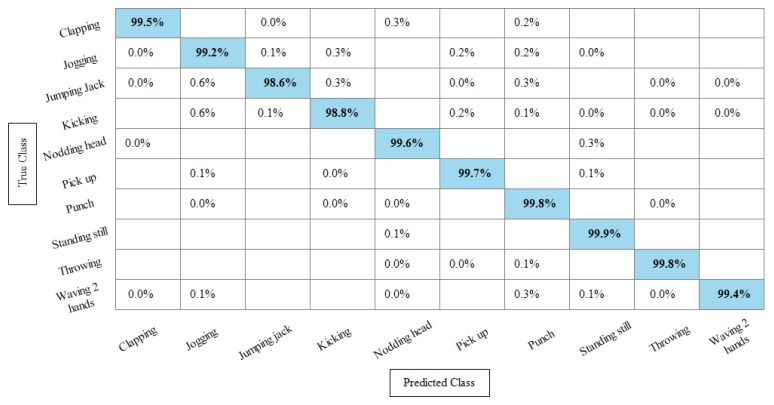
TPR-based confusion matrix of Cubic SVM after best feature selection using KcWKNN.

**Figure 15 sensors-21-07941-f015:**
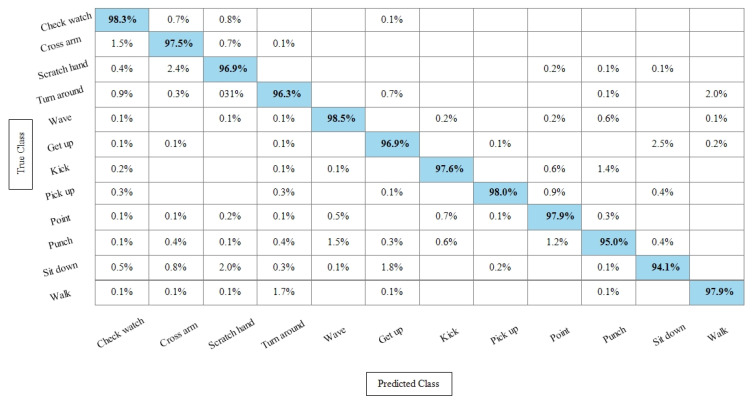
TPR-based confusion matrix of Fine KNN after best feature selection using KcWKNN.

**Figure 16 sensors-21-07941-f016:**
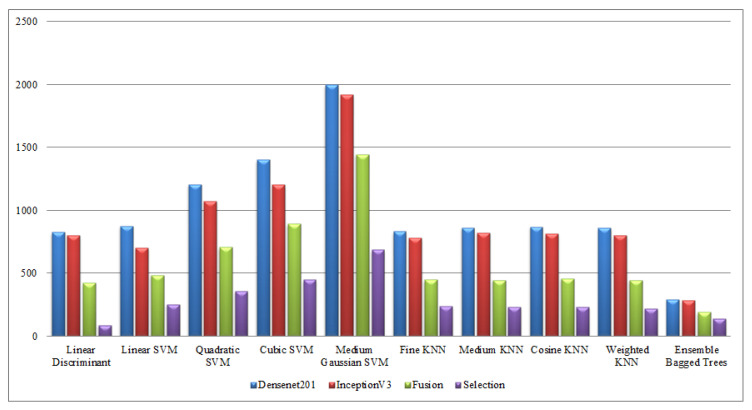
Computational time-based comparison of middle steps on KTH dataset.

**Figure 17 sensors-21-07941-f017:**
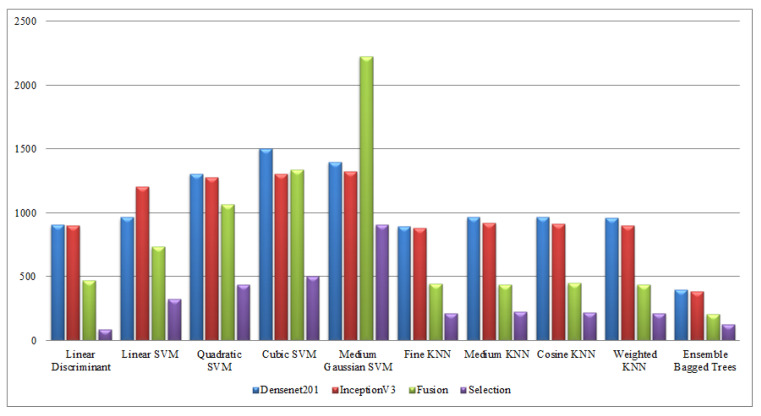
Computational time-based comparison of middle steps on Hollywood dataset.

**Figure 18 sensors-21-07941-f018:**
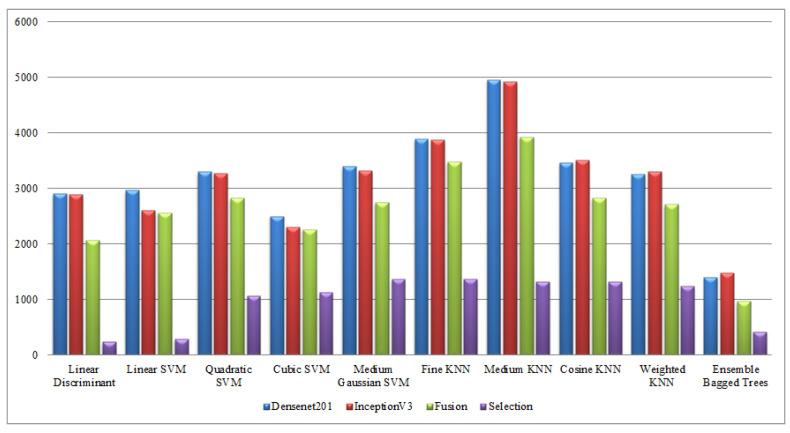
Computational time-based comparison of middle steps on WVU dataset.

**Figure 19 sensors-21-07941-f019:**
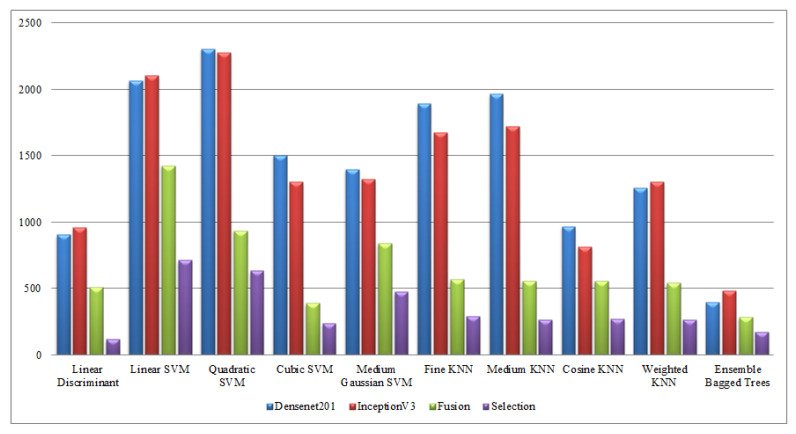
Computational time-based comparison of middle steps on IXMAS dataset.

**Table 1 sensors-21-07941-t001:** Classification accuracy on specific DenseNet201 deep model. The bold represents the best obtained values.

Classifier	Datasets Accuracy on DenseNet201 Deep Model			
	KTH	Hollywood	WVU	IXAMAS
Linear Discriminant	98.8	99.6	98.3	92.1
Linear SVM	98.0	98.3	97.1	86.6
Quadratic SVM	98.9	99.6	99.7	96.4
**Cubic SVM**	**99.3**	99.8	**99.8**	95.4
Medium Gaussian SVM	98.6	99.5	97.8	93.1
Fine KNN	98.7	**99.9**	99.3	**97.3**
Medium KNN	96.7	98.8	97.3	88.0
Cosine KNN	96.9	98.8	97.4	88.3
Weighted KNN	97.2	99.7	98.0	92.9
Ensemble Bagged Trees	89.6	98.2	94.5	82.9

**Table 2 sensors-21-07941-t002:** Classification accuracy on specific InceptionV3 deep model. The bold represents the best obtained values.

Classifier	Datasets Accuracy on DenseNet201 Deep Model			
	KTH	Hollywood	WVU	IXAMAS
Linear Discriminant	96.6	98.8	96.5	87.3
Linear SVM	95.4	96.3	93.5	81.3
Quadratic SVM	97.6	99.3	99.0	92.1
**Cubic SVM**	**98.1**	99.5	**99.1**	93.6
Medium Gaussian SVM	97.0	99.3	97.7	91.2
Fine KNN	97.6	**99.8**	98.4	**96.0**
Medium KNN	95.00	98.1	94.8	83.8
Cosine KNN	95.6	98.5	95.1	84.7
Weighted KNN	95.9	99.1	95.8	90.0
Ensemble Bagged Trees	89.0	92.4	90.5	73.3

**Table 3 sensors-21-07941-t003:** Achieved results on KTH dataset after fusion of deep features using SbE approach. The bold represents the best obtained values.

Classifier	Recall Rate (%)	Precision Rate (%)	FNR	Time (s)	F1 Score (%)	Accuracy (%)
Linear Discriminant	99.200	99.300	0.80	424.10	99.249	99.2
Linear SVM	98.400	98.616	1.60	487.10	98.508	98.4
Quadratic SVM	99.150	98.283	0.85	706.56	98.714	99.2
**Cubic SVM**	**99.300**	**99.433**	**0.70**	893.23	**99.366**	**99.3**
Medium Gaussian SVM	98.916	99.083	1.08	1445.8	98.999	98.9
Fine KNN	99.083	99.216	0.91	450.55	99.149	99.1
Medium KNN	96.700	97.233	3.30	447.37	96.965	96.8
Cosine KNN	97.516	97.716	2.48	459.33	97.616	97.5
Weighted KNN	97.483	97.916	2.51	447.59	97.699	97.6
Ensemble Bagged Trees	94.233	94.733	5.76	**192.96**	94.482	94.3

**Table 4 sensors-21-07941-t004:** Achieved results on Hollywood dataset after fusion of deep features using SbE approach. The bold represents the best obtained values.

Classifier	Recall Rate (%)	Precision Rate (%)	FNR	Time (s)	F1 Score (%)	Accuracy (%)
Linear Discriminant	99.775	99.825	0.22	469.75	99.800	99.9
Linear SVM	99.887	99.25	1.11	734.42	99.567	99.2
Quadratic SVM	99.550	99.725	0.45	1065.4	99.637	99.7
Cubic SVM	99.575	99.775	0.42	1337.4	99.674	99.8
Medium Gaussian SVM	99.287	99.675	0.71	2227.1	99.480	99.7
**Fine KNN**	**99.182**	**99.837**	**0.18**	**447.76**	**99.508**	**99.9**
Medium KNN	98.500	99.0125	1.50	437.47	98.755	99.1
Cosine KNN	99.037	98.975	0.96	449.13	99.006	99.3
Weighted KNN	99.250	99.45	0.75	439.29	99.349	99.6
Ensemble Bagged Trees	94.425	97.562	5.57	209.63	95.968	96.7

**Table 5 sensors-21-07941-t005:** Achieved results on WVU dataset after fusion of deep features using SbE approach. The bold represents the best obtained values.

Classifier	Recall Rate (%)	Precision Rate (%)	FNR (%)	Time (s)	F1 Score (%)	Accuracy (%)
**Linear Discriminant**	**99.79**	**99.78**	**0.21**	**2073.1**	**99.785**	**99.8**
Linear SVM	97.74	97.77	2.26	2567.7	97.755	97.7
Quadratic SVM	99.56	99.56	0.44	2824.5	99.560	99.6
Cubic SVM	99.56	99.57	0.44	2267	99.565	99.6
Medium Gaussian SVM	98.56	98.34	1.66	2749	98.449	98.3
Fine KNN	97.0	97.03	3.00	3486	97.015	97.0
Medium KNN	87.15	88.34	12.8	3933.5	87.741	87.2
Cosine KNN	87.98	89.01	12.1	2825.4	88.492	88.0
Weighted KNN	90.89	91.51	9.11	2716.7	91.198	90.9
Ensemble Bagged Trees	94.08	94.12	5.92	965.78	94.100	94.1

**Table 6 sensors-21-07941-t006:** Achieved results on IXMAS dataset after fusion of deep features using SbE approach. The bold represents the best obtained values.

Classifier	Recall Rate (%)	Precision Rate (%)	FNR (%)	Time (s)	F1 Score (%)	Accuracy (%)
Linear Discriminant	96.460	96.310	3.54	508.35	96.384	96.5
Linear SVM	91.030	91.230	8.97	1428	91.129	91.3
Quadratic SVM	96.670	96.680	3.33	936.8	96.675	96.7
Cubic SVM	97.216	97.225	2.78	390.9	97.220	97.3
Medium Gaussian SVM	96.016	96.066	3.98	840.3	96.041	96.1
**Fine KNN**	**97.180**	**97.250**	**2.82**	**570.56**	**97.215**	**97.4**
Medium KNN	88.360	88.890	11.6	560.06	88.624	88.9
Cosine KNN	89.141	89.516	10.8	559.83	89.328	89.7
Weighted KNN	92.475	92.625	7.52	543.5	92.549	92.8
Ensemble Bagged Trees	80.291	81.550	19.7	284.31	80.915	81.4

**Table 7 sensors-21-07941-t007:** Achieved results on KTH dataset after best feature selection using KcWKNN. The bold represents the best obtained values.

Classifier	Recall Rate (%)	Precision Rate (%)	FNR (%)	Time (s)	F1 Score (%)	Accuracy (%)
Linear Discriminant	98.080	98.516	1.92	87.805	98.297	98.1
Linear SVM	97.633	97.933	2.36	255.42	97.783	97.7
Quadratic SVM	98.600	98.866	1.40	360.10	98.733	98.7
**Cubic SVM**	**98.916**	**99.116**	**1.09**	**451.40**	**99.016**	**99.0**
Medium Gaussian SVM	98.2833	98.483	1.71	687.37	98.383	98.3
Fine KNN	98.616	98.833	1.38	237.93	98.724	98.7
Medium KNN	95.483	96.366	4.51	231.39	95.922	95.7
Cosine KNN	97.016	97.183	2.98	230.18	97.099	97.0
Weighted KNN	96.233	97.000	3.76	222.90	96.615	96.4
Ensemble Bagged Trees	94.150	93.716	5.8	140.57	93.632	94.2

**Table 8 sensors-21-07941-t008:** Achieved results on Hollywood dataset after best feature selection using KcWKNN. The bold represents the best obtained values.

Classifier	Recall Rate (%)	Precision Rate (%)	FNR (%)	Time (s)	F1 Score (%)	Accuracy (%)
Linear Discriminant	99.087	99.450	0.912	88.375	99.268	99.4
Linear SVM	97.937	98.687	2.062	323.99	98.311	98.6
Quadratic SVM	99.262	99.587	0.737	439.41	99.424	99.5
Cubic SVM	99.387	99.675	0.612	501.67	99.531	99.7
Medium Gaussian SVM	98.587	99.500	1.412	910.78	99.041	99.5
**Fine KNN**	**99.812**	**99.837**	**0.187**	**213.33**	**99.825**	**99.8**
Medium KNN	97.225	98.550	2.775	224.52	97.883	98.5
Cosine KNN	98.325	98.862	1.675	221.19	98.593	98.9
Weighted KNN	98.575	99.412	1.425	215.89	98.992	99.2
Ensemble Bagged Trees	87.050	94.287	12.95	126.72	90.524	97.7

**Table 9 sensors-21-07941-t009:** Achieved results on WVU dataset after best feature selection using KcWKNN. The bold represents the best obtained values.

Classifier	Recall Rate (%)	Precision Rate (%)	FNR (%)	Time (s)	F1 Score (%)	Accuracy (%)
Linear Discriminant	98.50	98.53	1.50	241.48	98.515	98.5
Linear SVM	96.51	96.57	3.49	293.2	96.539	96.5
**Quadratic SVM**	**99.37**	**99.38**	**0.63**	**1064.6**	**99.375**	**99.4**
Cubic SVM	99.43	99.44	0.57	1124.0	99.435	99.4
Medium Gaussian SVM	98.24	98.25	1.76	1363.7	98.245	98.2
Fine KNN	96.55	96.59	3.45	1365.1	96.570	96.5
Medium KNN	86.80	87.98	13.2	1322.0	87.386	86.8
Cosine KNN	87.61	88.73	12.39	1316.2	88.166	87.6
Weighted KNN	90.33	91.07	9.67	1236.8	90.698	90.3
Ensemble Bagged Trees	94.71	94.75	5.29	423.37	94.730	95.7

**Table 10 sensors-21-07941-t010:** Achieved results on IXMAS dataset after best feature selection using KcWKNN. The bold represents the best obtained values.

Classifier	Recall Rate (%)	Precision Rate (%)	FNR (%)	Time (s)	F1 Score (%)	Accuracy (%)
Linear Discriminant	91.583	91.516	8.41	119.8	91.549	91.7
Linear SVM	88.050	88.400	11.95	714.13	88.224	88.5
Quadratic SVM	95.008	95.083	4.99	634.7	95.045	95.1
Cubic SVM	95.783	95.866	4.21	239.4	95.824	95.9
Medium Gaussian SVM	94.466	94.933	5.53	475.5	94.699	94.6
**Fine KNN**	**97.075**	**96.991**	**2.92**	**290.69**	**97.033**	**97.1**
Medium KNN	86.383	86.925	13.61	266.24	86.653	86.9
Cosine KNN	88.066	88.233	11.93	270.74	88.149	88.5
Weighted KNN	90.975	91.966	9.02	263.74	91.468	91.2
Ensemble Bagged Trees	83.433	85.108	16.5	175.78	84.261	84.8

**Table 11 sensors-21-07941-t011:** Comparison of the proposed design with existing techniques in terms of accuracy. The bold represents the best obtained values.

Reference	Dataset	Accuracy (%)
Muhammad et al. [45], 2020	KTH	98.30
**Proposed method**	**KTH**	**99.00**
Muhammad et al. [4], 2020	IXMAS	95.20
Amir et al. [55], 2021	IXMAS	87.48
**Proposed method**	**IXMAS**	**97.10**
Muhammad et al. [56], 2020	WVU	99.10
Muhammad et al. [57], 2019	WVU	99.90
**Proposed method**	**WVU**	**99.40**
Evan et al. [58], 2008	Hollywood	91.80
**Proposed method**	**Hollywood**	**99.20**

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
