# Peer review of "Human Action Recognition: A Paradigm of Best Deep Learning Features Selection and Serial Based Extended Fusion"

_sensors, 2021, doi:10.3390/s21237941_

Round 1

Reviewer 1 Report

This manuscript deals with an important and timely topic: automatic detection of human actions (HAR). With more and more wearables and more data about human activity, there is a bounty of data processing that could lead to new insights into human activity. However, this has to be performed automatically and machine learning (ML) happens to be the only way to address this challenge.

In this manuscript, the authors deal with this issue in the particular case of a data-rich video data. 

The developed framework is interesting and the methodology sound. One concern I have is the literature review. It seems rather shallow and the authors are missing some important review papers recently published in MDPI Sensors: e.g. 

Manivannan et al. "On the challenges and potential of using barometric sensors to track human activity." Sensors 20.23 (2020): 6786.

Another issue that I would like to see addressed is that of the confusion matrices. Most of these confusion matrices presented are sparse, and information along the main diagonal is presented. However, to truly assess the performance of the ML approach, it is critical to have the values for non-diagonal entries too.

I hope the authors can address these issues in a revision.

Author Response

Response sheet has been attached. thanks

Reviewer 2 Report

In this research work, the authors propose a design based on a deep learning approach to recognise human behaviour. The main steps include feature mapping, feature fusion and feature selection. In the initial feature mapping step, the authors consider two pre-trained models, DenseNet201 and InceptionV3. Then, the extracted deep features are fused using a sequence-based extension (SbE) approach. Subsequently, the best features were selected using a kurtosis-controlled weighted KNN. The selected features were classified using several supervised learning algorithms.

The main contribution of the authors is the proposed sequence-based extension (SbE) method for fusion of extracted depth features, using kurtosis-controlled weighted KNN to select the best features. There is an improvement in recognition accuracy and speed.

I suggest this article needs a major revision.

Main issues and amendments:

  1. The author mentions in the main contribution“Selected two deep learning models based on their individual image classification performance and re-trained these models on target action recognition datasets us-ing transfer learning”. But, I have the feeling that this is not an author's contribution and that the pre-training was done using proven methods. It is suggested that the authors re-evaluate the formulation of this contribution.
  2. The authors chose the ratio of the fourth order moment to the fourth power of the variance as the threshold, but why this metric was chosen is not explained, nor is the process of comparison. It is recommended that the appropriate additions be made.
  3. The authors conclude that using this method improves computational accuracy and reduces computational time, but doing feature fusion and feature selection increases the complexity increasing the cost and running time, so overall, there is no clear account of whether the running time is increasing or decreasing. It is recommended that the authors provide further clarification in terms of accuracy and efficiency

Suggestions for minor changes:

  1. Line 93: Proposed a feature selection technique name(named) Kurtosis controlled Weighted KNN 93 (KcWKNN).
  2. line98:The rest of the manuscript is organizing as follows: Related(related) work is presenting in Section
  3. line246:maximum epochs are (epoch is) 100
  4. line251:Figure 5 represent(represents) DenseNet201 modified model.
  5. line270:The fused vectors are combined in a 269 single feature vector and obtained(to obtain) a feature vector
  6. line274行:having(has)
  7. line278:The formula has an incorrect corner marker
  8. line305,308,311:Missing or incorrectly labelled formulae
  9. Figure1 is too small to read in the column indicating DenseNet 201
  10. Figure 5 linking arrows are not in the same format as other tables

Author Response

(The authors gave the same response as above.)

Round 2

Reviewer 2 Report

The previous questions were basically responded to or answered.